# Are Fruit Juices Healthier Than Sugar-Sweetened Beverages? A Review

**DOI:** 10.3390/nu11051006

**Published:** 2019-05-02

**Authors:** Alexandra Pepin, Kimber L. Stanhope, Pascal Imbeault

**Affiliations:** 1Behavioral and Metabolic Research Unit, School of Human Kinetics, Faculty of Health Sciences, University of Ottawa, Ottawa, ON K1N6N5, Canada; alexandra.pepin@uottawa.ca; 2Department of Molecular Biosciences, School of Veterinary Medicine, University of California, Davis, CA 95616, USA; 3Institut du savoir Montfort, Hôpital Montfort, Ottawa, ON K1K0T2, Canada

**Keywords:** free sugars, fruit juices, fructose, high-fructose corn syrup, sugar-sweetened beverages, dyslipidemia

## Abstract

Free sugars overconsumption is associated with an increased prevalence of risk factors for metabolic diseases such as the alteration of the blood lipid levels. Natural fruit juices have a free sugar composition quite similar to that of sugar-sweetened beverages. Thus, could fruit juice consumption lead to the same adverse effects on health as sweetened beverages? We attempted to answer this question by reviewing the available evidence on the health effects of both sugar-sweetened beverages and natural fruit juices. We determined that, despite the similarity of fruits juices to sugar-sweetened beverages in terms of free sugars content, it remains unclear whether they lead to the same metabolic consequences if consumed in equal dose. Important discrepancies between studies, such as type of fruit juice, dose, duration, study design, and measured outcomes, make it impossible to provide evidence-based public recommendations as to whether the consumption of fruit juices alters the blood lipid profile. More randomized controlled trials comparing the metabolic effects of fruit juice and sugar-sweetened beverage consumption are needed to shape accurate public health guidelines on the variety and quantity of free sugars in our diet that would help to prevent the development of obesity and related health problems.

## 1. Introduction

The ingestion of free sugars may favor the overconsumption of energy, thus promoting the development of risk factors associated with metabolic diseases such as hypertriglyceridemia, hypercholesterolemia, and insulin resistance [1,2,3]. Moreover, the current literature strongly suggests that ingestion of sugar-sweetened beverages increases the cardiometabolic risk and risk factors more than isocaloric amounts of complex carbohydrates [4]. Free sugars are defined as any types of simple sugars (monosaccharides or disaccharides) that have been added to beverages or food products during their transformation or preparation by food industries or by the consumer per se, plus sugars naturally present in fruit juices, fruit juice concentrates, honey, and various syrups [5]. Sugars that occur in the natural structure of entire fruits and vegetables as well as those from milk (lactose) are not categorized as free sugars [5,6]. Added sugars include sugars and syrups that are added during the preparation or the transformation of food and beverages. Therefore, natural fruit juices do not contain added sugars, but on the basis of the above definition, they are a significant source of free sugars.

While it is mainly accepted that the overconsumption of sugar-sweetened beverages may lead to adverse effects on health [3,4], the evidence pertaining to the consumption of fruit juices is a matter of debate. This is reflected in the inconsistency between dietary guidelines that relate to the consumption of natural fruit juice. The World Health Organization (WHO) recommends reducing the intake of free sugars to less than 10% (and, ideally, less than 5%) of total daily energy intake, thus including sugars naturally present in fruit juices in the category of sugars whose consumption should be reduced. The 2015–2020 Edition of the Dietary Guidelines for Americans, a resource for health professionals and policymakers for the design and implementation of nutrition programs in the United States, recommends consuming less than 10% of calories per day from added sugars, thus not including sugars naturally present in fruit juices in the category of sugars whose consumption should be reduced [7]. However, the American Academy of Pediatrics recommends limiting the consumption of fruit juice for children between the age of 1 and 3 years to 4 oz (120 mL)/day, for those from 4 to 6 years to 4–6 oz (120–180 mL)/day, and for those from 7 to 18 years to 8 oz (240 mL)/day [8,9].

There are currently no data estimating the average consumption of free sugars in the United States. Still, fruit juices are commonly consumed on a daily basis by many American adults and children. This review aims to summarize the effects of free sugar consumption, especially from sugar-sweetened beverages, on metabolic health and to consider whether fruit juices may lead to similar health outcomes.

## 2. Added Sugars: Sucrose and High-Fructose Corn Syrup

Simple sugars have been a part of the human diet for millennia. They were provided mainly by fruits and honey until white sugar (sucrose) became a common consumer product in the 19th century [10]. Nowadays, the worldwide consumption of sucrose is widespread, to the extent that it has tripled over the past 50 years [11]. In the United States, 77% of all calories purchased from 2005 to 2009 contained sweeteners, of which corn syrup, cane sugar, High-Fructose Corn Syrup (HFCS), and fruit juice concentrate were listed as the most commonly used [12]. Sucrose is naturally occurring in sugar cane and sugar beet and therefore extracted and purified directly from sugar cane or beet sap. In contrast, HFCS, which replaces sucrose in 40% of processed foods and beverages, is not naturally occurring [13].

In 1864, the Union Sugar Company generated corn syrup for the first time by treating cornstarch with enzymes to break down the complex carbohydrate into glucose and glucose polymers [14]. Although corn syrup was more affordable than basic cane or beet sugar, it tasted less than half as sweet as sucrose. The isomerization of corn glucose to fructose using microbial enzymes was discovered in the late 1950s by the Clinton Corn Processing Company [14]. This product was, however, not economically viable at that point, mainly because of its instability. Yoshiyuki Takasaki, a Japanese scientist from the Japanese Agency of Industrial Science and Technology, was the first to develop HFCS by isolating a heat-stable glucose isomerase (xylose isomerase) derived from *Streptomyces* sp. in the late 1960s [14]. HFCS can be labeled under many different appellations such as isoglucose or glucose-fructose. The increasing popularity of HFCS amongst food industries was principally due to a more favorable price (the USA being the world’s largest producer of corn) and to the fact that its liquid form both inhibits crystallization and helps to maintain the moisture content of baked goods [15], while also being easily diluted in sugar-sweetened beverages. A few years later, HFCS-55 (45% glucose, 55% fructose) was marketed, and HFCS largely replaced sucrose in sugary drinks made by major brands of cola beverages (1984) [14]. Amongst 43 countries from which the HFCS consumption per capita was analyzed in 2000, 2004, and 2007, the United States was ranked the highest with a yearly HFCS consumption of 24.78 kg/per capita [16]. Nevertheless, HFCS consumption has been steadily decreasing since 2009 in the United States, as opposed to sucrose consumption which tends to increase [17]. This observed shift in sweetener choice by the food industry is probably due to the fact that HFCS consumption has been badly publicized in the public media in recent years. The 2017 per capita supply of caloric sweeteners in the United States was estimated to be 158 g [17], of which 86 g was derived from sucrose (cane or beet sugar), 50 g from HFCS, and 22 g from glucose corn syrup, dextrose, honey, and other syrups [17]. The actual intake can be estimated by assuming losses due to spoilage and wastage of 11% at the retail and institutional level and of 20% by the consumer [18]. 

The sucrose molecule is composed of one unit of glucose covalently bonded to one unit of fructose, the fructose concentration of sucrose therefore being 50%. In contrast, glucose and fructose occur in their free forms in HFCS [19]. This makes it possible to vary the fructose-to-glucose ratio in the syrup mixture and create different formulations. According to the Corn Refiners Association, there are two primary compositions of HFCS which are HFCS-42 and HFCS-55, respectively produced with 42% and 55% of fructose [20]. HFCS-55 is considered as the most commonly used form amongst food industries, but there are currently no obligations for manufacturers to disclose what formulation of HFCS they use on the ingredient list of the food labels. Thus, the fructose content of most processed foods and sugar-sweetened beverages remains uncertain [21]. Ventura et al. (2011) measured the sugar content of 23 popular sugar-sweetened beverages purchased at the grocery store with a focus on fructose. The results from this study revealed that 15 of those beverages had a fructose-to-glucose ratio exceeding 55%, with a mean fructose content of 59%. Moreover, several renown-brand beverages appeared to be made with a formulation of HFCS containing 65% of fructose instead of 55% [21]. However, this study was criticized for the methods used, which were designed to identify sucrose, glucose, and fructose, but were not sensitive enough to precisely measure other traces of sugars (such as maltose) if they were present in the beverages [22]. In response, a study funded by the International Society of Beverage Technologists randomly selected 80 beverages known to be sweetened with HFCS-55 by Coca-Cola Company, PepsiCo, and Dr Pepper Snapple Group in order to validate the fructose content. The mean fructose content of those beverages resulted to be 55.58%, with a 95% confidence interval of 55.51–55.65% [23]. It should be noted that this study sampled amongst beverages known to be sweetened with HFCS-55 only, thereby not including the ones that are potentially sweetened with HFCS-65, such as the ones distributed in the Coca-Cola Freestyle dispensers [24]. Walker et al. (2014) [25] selected 10 of the 23 sweetened beverages (based on popularity) that were previously analyzed by Ventura and colleagues (2011) [21] for further analysis using two alternative methods to determine the fructose content. In beverages listing HFCS as an ingredient, the mean fructose concentration was 59.4 ± 8.9 g/L, which corroborates the results of Ventura et al. (2011) [21]. Three of the selected beverages had a free fructose concentration exceeding 60 g/L. Maltose was detected in 8 of the 10 beverages, and its concentration was less than 2%, whereas galactose and lactose were not detected in any of the beverages [25]. The results of these two studies [21,25] provide evidence that many popular beverages sweetened with HFCS have a fructose content greater than 55%, thus suggesting the use of a higher concentration of HFCS. From a biological standpoint, it is unlikely that a meaningful difference exists between sucrose and HFCS-55 based on their fructose content. However, there may be a significant biological difference between fructose-to-glucose ratios of 50:50 and 60:40. Thus, the use of HFCS at a fructose concentration of more than 55% by the sweetened beverage industries, without any obligation to disclose the formulation to consumers, is concerning. 

## 3. How Does Fructose Can Alter Lipemia?

### 3.1. Glucose Metabolism

Concerns about the ratio of fructose to glucose in beverages relate to the well-established differences between glucose and fructose metabolism. Glucose enters the enterocytes mostly by secondary active transport via sodium–glucose transporters (SGLT1) located in the apical membrane of the enterocytes. SGLT1 transporters have a high-affinity for glucose, but a low transportation capacity. Thus, under high concentrations of glucose in the lumen of the intestine, glucose also enters the enterocytes by facilitated diffusion via low-affinity, but high-capacity glucose transporters (GLUT2) [26]. GLUT2 transporters are expressed to a lesser extent in the apical membrane of the enterocytes but can be rapidly translocated from the basolateral membrane to the apical membrane to enhance glucose uptake under high concentrations of intestinal glucose [26]. Then, glucose exits the enterocytes to enter the bloodstream by facilitated diffusion via GLUT2 transporters located in the basolateral membrane (Figure 1). Glucose is then transported to the liver by the portal vein. Hepatic glucose metabolism is regulated by insulin and hepatic energy needs. This allows much of ingested glucose arriving via the portal vein to bypass the energy-replete liver and to rapidly reach the systemic circulation [27]. The first step of glycolysis, the responsible pathway for glucose metabolism, is the phosphorylation of glucose on its 6th carbon by the enzyme glucokinase (hexokinase). This step is then followed by an isomerization reaction resulting in fructose 6-phosphate (F6P). The major limiting step of glycolysis is the phosphorylation of F6P to fructose 1,6-biphosphate (catalyzed by phosphofructokinase), which will allow the molecule to be cleaved in two three-carbon units that can later be used to generate ATP [28].

### 3.2. Fructose Metabolism

When fructose is ingested, it enters enterocytes through a specific fructose transporter (GLUT5), which is independent of sodium-glucose linked transporters and does not require ATP hydrolysis as opposed to SGLT1 [1]. Fructose will then enter the systematic circulation in a similar way to glucose, that is by facilitated diffusion via GLUT2 transporters (Figure 1). A small part of dietary fructose will be converted and released in the bloodstream by the enterocytes as glucose [1,29], lactate [1,29], and fatty acids in chylomicrons [30,31,32,33,34] (Figure 2). Yet, the role of the enterocytes in determining the metabolic fate of fructose has not been clearly established. 

The excessive amount of fructose will spillover to the liver where it is nearly all cleared from the portal blood after its first pass. Fructose will be rapidly phosphorylated (catalyzed by fructokinase C, a key enzyme in the metabolism of fructose [36,37,38]) on its first carbon, resulting in fructose 1-phosphate (F 1-P) instead of F6P [28] (Figure 2). F 1-P has the capacity to bypass the first limiting step of glycolysis [28] without being regulated by insulin nor inhibited by ATP production. F 1-P will mostly be metabolized by aldolase B into glyceraldehyde 3-P (G 3-P). Subsequently, G 3-P can be: (1) converted into pyruvate (resulting in acetyl-CoA production); (2) converted and released as lactate; or (3) converted to glucose (gluconeogenesis) [1]. The largest part of G 3-P will be converted to glucose, which can be stored as glycogen or released as glucose 6-phosphate in the bloodstream [35]. In fact, in the 1990s, isotope tracing with intravenously infused ^13^C-labeled fructose in humans showed that ~50% of a fructose load was converted and recirculated as ^13^C-labeled glucose [39]. When released in the systematic circulation, glucose and lactate can be utilized as an energy substrate by the brain, heart, and muscle tissue [35] (Figure 2). As mentioned previously, a large part of F 1-P will be metabolized in G 3-P. Nonetheless, the excessive supply of fructose spilled to the liver has also been shown to simultaneously inhibit lipid oxidation [40] and to enhance hepatic de novo lipogenesis (DNL) [41,42]. DNL has the capacity to convert fructose, more precisely F 1-P, into fatty acids [43], thus consequently increasing the intrahepatic lipid supply. Elevated levels of intrahepatic lipids content favor very low-density lipoproteins (VLDL) production and secretion [44], which also leads to increased levels of postprandial triglycerides and dyslipidemia [1,42] (Figure 2). Increased levels of intrahepatic lipids are associated with hepatic insulin resistance [45,46]. Of note, a systematic review indicated that fructose consumption in an energy-matched exchange for other carbohydrates (mostly glucose) induces hepatic insulin resistance [47]. This suggests that the promotion of hepatic insulin resistance by fructose could not be attributed only to the excess of energy intake under hypercaloric diet conditions. Knowing that DNL is more strongly activated in the insulin-resistant liver [48], fructose consumption has the potential to generate a vicious cycle that would further increase the intrahepatic lipid supply, thus amplifying VLDL-triglyceride production and secretion [2]. Continued exposure to triglycerides promotes muscle lipid accumulation [49], which may also promote whole-body insulin resistance [50,51].

## 4. Sugar-Sweetened Beverages and Their Implication in Metabolic Health

### Free Sugars Consumed from Sugar-Sweetened Beverages Induce Metabolic Perturbations

Epidemiological studies have shown that the consumption of sugar-sweetened beverages is associated with increased energy intake, long-term weight gain, and prevalence of metabolic and cardiovascular diseases [3,52,53]. Experimental evidence strongly suggests the fructose component of HFCS and sucrose promotes metabolic perturbations such as dyslipidemias [41,42,49,54,55,56,57,58,59,60,61,62,63,64] and insulin resistance [35,42,57,65]. The consumption of free sugars at the level that is currently consumed by Americans may adversely alter lipemia. It has indeed been shown that sucrose, when consumed at 13% of estimated daily energy requirements (Ereq) (80 g/day) as a beverage along with a usual ad libitum diet for three weeks, led to increased low-density lipoprotein cholesterol (LDL-C) and decreased hepatic insulin sensitivity in healthy young men compared to the consumption of glucose [65]. It should be noted that daily energy requirements refer to the calorie intake needed to balance energy expenditure in order to maintain a healthy individual’s body weight stable. These results are in line with recent findings showing that a 12-week intervention where 13% of diet energy as fructose was served in the habitual diet of 71 men with abdominal obesity led to enhanced DNL and increased body weight, liver fat content, and postprandial triglyceride levels [56]. Young men and women consuming beverages containing 0, 10, 17.5, and 25% of Ereq as HFCS along with ad libitum diets exhibited a dose-dependent increase of LDL-C, apolipoprotein B (the protein backbone of VLDL), apoCIII, uric acid, and postprandial triglycerides [66]. Importantly, even the group consuming the 10% dose exhibited significantly increased concentration of LDL-C, apolipoprotein B, and postprandial triglycerides compared with their baseline concentration [66]. In a six-month dietary intervention study, subjects consuming one liter of sucrose-sweetened beverages/day exhibited increased triglycerides, cholesterol, and liver fat [49]. A recent meta-analysis revealed that the dose and overall caloric intake of free sugars have the strongest deteriorating effect on blood lipids as compared to interventions where isocaloric substitution of free sugars with complex carbohydrates is provided [67]. However, most of these studies are limited in that free sugars were consumed along with the subjects’ own usual ad libitum diets during most of the intervention, thus the total amount of free sugars consumed daily are unknown. This prevents attributing adverse effects to precise doses of free sugars. Another limitation to the above studies is that many of the subjects exhibited increases in body weight. This makes it difficult to separate the direct effects of fructose from those indirectly mediated by increased adiposity.

There are several studies that compared sugar (sucrose, HFCS, and/or fructose) consumption from sugar-sweetened beverages with isocaloric substitutions for complex carbohydrates [41,57] or glucose [59,65], or from low-sugar diets [54,55,61,66,68,69] in healthy individuals on health outcomes (Figure 3 and Table 1 summarizes these studies and their effects on fasting triglycerides and fasting LDL-C levels). However, to our knowledge, there are only three studies [41,68,69] in which the effects of sugar consumption from sugar-sweetened beverages at levels less than 30% Ereq were investigated in healthy subjects utilizing a controlled dietary protocol that prevented body weight gain (eucaloric) and diet macronutrient variations between experimental groups. Black et al. (2006) [69] conducted a six-week crossover study with healthy men who consumed eucaloric diets containing high (25% Ereq) or low (10% Ereq) amounts of sucrose. The 25% Ereq sucrose diet increased the levels of total cholesterol by 15% and of LDL-C by 24% as compared to the 10% Ereq sucrose diet; the authors suggested this could have been caused by the high-sucrose diet containing more saturated fats [69]. More recently, Lewis et al. (2013) [68] conducted a randomized six-week crossover study in which individuals with obesity consumed low- (5%) or high- (15%) sucrose diets for six weeks. While no differences in lipid levels were observed, the subjects displayed increased glucose (5.0 ± 0.2 vs. 5.4 ± 0.2 mmol/L, *p* < 0.01) and insulin responses (59.0 vs. 109.2 mU/L, *p* < 0.01) during the oral glucose tolerance test (OGTT) when consuming the high-sucrose diet [68]. Likewise, Schwarz and colleagues (2015) [41] investigated eight healthy men consuming crossover diets containing eucaloric amounts of either fructose or complex carbohydrates (25% of Ereq) for nine days, while maintaining their body weight stable and providing the same macronutrient distributions. The subjects exhibited elevated DNL (average, 18.6 ± 1.4% vs 11.0 ± 1.4%; *p* = 0.001), liver lipids (median, + 137%, *p* = 0.016), and postprandial triglycerides (in seven of eight participants: average, 172 ± 29 vs. 140 ± 28 mg/dL; *p* = 0.002) when consuming the high-fructose diet compared with the complex-carbohydrate diet [41]. Overall, these results suggest that sucrose or fructose consumption at levels as low as 18% Ereq, increases the risk factors for metabolic diseases, even when consumed with a diet that does not promote weight gain. However, this conclusion could be confounded by the vulnerability of subjects with obesity [68], the different saturated fat content between diets [69], and the use of pure fructose instead of HFCS or sucrose [41]. Therefore, because of the limited number of studies and the limitations of these studies, it is difficult to establish firm conclusions regarding the weight-independent effects of sugar consumption on health outcomes.

## 5. What about the Effects of Natural Fruit Juices on Blood Lipids?

It is well accepted that fruit intake is protective for human health, but there is no clear consensus about the effects of consuming the juices that are extracted from them [70]. The primary component of fruit juices, apart from water, is free sugars in a concentration of about 100–120 g/L depending on the variety and the quality of fruits [71]. The fructose content of most natural fruit juices is quite similar to that of beverages sweetened with HFCS-55. For instance, orange juice has an average total fructose concentration (including free fructose and fructose from sucrose) of 51–57 g/L, which represents 52 to 54% of its total sugar content [25,72]. However, it is important to note that, despite their similarity to sugar-sweetened beverages in terms of fructose composition, fruit juices are also a source of various bioactive compounds such as phytonutrients, whose consumption has been shown to be beneficial to human health and the prevention of chronic diseases [73]. Given the comparable free sugar and fructose content of natural fruit juices and HFCS-55-sweetened beverages, could their consumption potentially lead to similar metabolic effects, despite their different composition in vitamins, minerals, and antioxidants? 

### 5.1. Population Studies

A meta-analysis and systematic review from 17 cohort studies examined the prospective associations between the consumption of different types of beverages containing free sugars such as sugar-sweetened beverages and fruit juices in subjects with type 2 diabetes [74]. The daily consumption of 250 mL of sugar-sweetened beverages was associated with an increased incidence risk of type 2 diabetes by 13% (95% confidence interval 6% to 21%, *I*^2^ for heterogeneity = 79%), whereas the same consumption of fruit juices elevated the incidence risk by 7% (1% to 14%, *I*^2^ = 51%), both independently of adiposity [74]. Another recent population study showed that the consumption of more than five servings/week (200 mL/serving) of either natural fruit juices or sugar-sweetened beverages was associated with an increased risk of metabolic syndrome, more specifically of abdominal obesity and hypertriglyceridemia [75]. Surprisingly, the consumption of 1–5 servings of natural fruit juices weekly (without any sugar-sweetened beverages) was inversely associated with the same risk factors [75]. Furthermore, the substitution of 250 mL of natural fruit juices per day by the same amount of water for four years was associated with an 8% abated risk of type 2 diabetes [76]. On the other hand, other cohort studies that reported an association of sugar-sweetened beverage consumption with the risks of chronic diseases, such as metabolic syndrome [77], cardiovascular diseases [78,79], and type 2 diabetes [80,81,82,83,84], failed to find the same association with natural fruit juices. One major limitation of population studies is that they rely on self-reported food intake data. Additionally, there are many confounding factors. For example, in three of the aforementioned studies, participants who reported consuming fruit juices on a daily basis also reported being more physically active than the non-consumers or the consumers of sugar-sweetened beverages [80] or were shown to have general healthier diets [83,84], which can both have an impactful effect on lipid metabolism and insulin resistance.

### 5.2. Intervention Studies

#### 5.2.1. Absence of Effect

A meta-analysis of 19 randomized controlled trials investigating the effects of various fruit juices, concentrated fruit juices, and fruit juice powders suggested that fruit juices had a borderline significant effect on the reduction of diastolic blood pressure and did not affect total or LDL-C [85]. However, nearly all these studies investigated a dose of free sugars provided by fruit juices well below 10% Ereq, whereas adverse effects in relation to the consumption of sugar-sweetened beverages and blood lipids have never been observed at doses below 10% Ereq. Moreover, very few studies included in the meta-analysis directly compared the effects of fruit juices to sugar-sweetened beverages. The interventions were all conducted with usual diets with or without restrictions of consuming other fruits juices, alcohol, food sources of antioxidants, and specific fruits and vegetables. None of the 19 studies included in the meta-analysis controlled for energy intake nor for macronutrient distribution. In a 12-week trial, Simpson et al. (2016) showed that the consumption of 250 mL/day of natural orange juice in overweight hypercholesterolemic men did not affect body weight, insulin sensitivity, and circulating lipids when compared to the consumption of energy-matched orange sugar-sweetened beverage [86]. Nevertheless, a 250 mL dose of fruit juice provides 25 g of total free sugars [25], which represents an average of only 5% Ereq.

#### 5.2.2. Detrimental Effects 

In a four-week dietary controlled crossover study, 23 healthy men and women ingested a daily intake of various apple supplementations, including 550 g of whole fresh apples or 500 mL of clear or cloudy apple juice [87]. The consumption of 500 mL/day of clear apple juice was shown to significantly increase LDL-C by 6.9% as compared to whole apples, despite no adverse effects on high- density lipoprotein cholesterol (HDL-C) and plasma triglycerides concentrations. Interestingly, elevation in LDL-C was not observed in the group consuming cloudy apple juice. The lack of soluble fibers (pectin) and lower polyphenols levels in clear apple juice compared to cloudy apple juice have been suggested to explain the different metabolic responses observed [87]. At a dose closer to 15% Ereq, Kurowska et al. (2000) provided 750 mL/day of orange juice for four weeks to 25 subjects (men and women) with hypercholesterolemia and observed a reduction of the LDL to HDL cholesterol ratio by 16% due to an elevation of HDL-C, without, however, affecting LDL-C levels [88]. While an elevation in HDL-C could be interpreted as beneficial for cardiovascular health, this intervention also led to a 30% increase in triglyceride levels (from 1.56 ± 0.72 to 2.03 ± 0.91 mmol/L; *p* < 0.02) without any weight gain being observed [88]. 

#### 5.2.3. Beneficial Effects

In contrast with the results reported by Kurowska et al. (2000) [88], Cesar et al. (2006) [89] showed that the effects of 750 mL orange juice combined with an ad libitum diet for 60 days in men with normal cholesterol levels non-significantly decreased LDL-C (−0.08 mmol/L). When the same dose was given to men with high cholesterol levels, a significant decrease of LDL-C (−0.47 mmol/L) was observed, while triglyceride levels remained unchanged in both groups [89]. More recently, 78 participants living with obesity were provided individualized weight-loss diet prescriptions that included randomization into two groups, one of which including 500 mL/day of orange juice. After 12 weeks, both groups exhibited similar body mass index (BMI) and fat mass loss. Subjects in the orange juice group showed a reduction of homeostasis model assessment-insulin resistance by 33% (*p* = 0.04), of LDL-C by 24% (*p* ≤ 0.001), and of the inflammation biomarker C-reactive protein by 33% (*p* = 0.001) compared with the placebo group [90].

A plausible explanation as to why fruit juices may lead to different health effects from those of sugar-sweetened beverages is that fruit juices are a source of various bioactive compounds, such as vitamins (vitamin C, folate, etc.), minerals (mostly potassium), and antioxidants [73], while sugar-sweetened beverages generally have none. Antioxidants were shown to decrease oxidative stress [91], which plays an important role in the prevention of cardiovascular disease, including atherosclerosis, diabetes, and dyslipidemia [92]. Hesperidin and naringenin are the most important citrus flavanones (type of antioxidant) present in orange juice [93], which is the most studied type of fruit juice due to its popularity amongst consumers. A limited number of human studies [94,95] suggest that the anti-inflammatory effects [96] induced by the ingestion of those flavanones could contribute to the prevention of chronic diseases and therefore potentially counteract the detrimental effects of free sugars when consumed in limited amounts [97]. It is also hypothesized that fruit juices may differently affect the gut microbiota as compared to sugar-sweetened beverages and therefore lead to different metabolic effects [98]. Notwithstanding, a recent meta-analysis concluded that there is weak evidence to suggest that blood lipid levels may be improved with consumption of fruit juices [99]. The authors indicate that factors such as the variety of natural fruit juices, the dose, the length of exposure, the study design (randomized vs cross-sectional studies), as well as the dietary and exercise patterns of participants must be considered in future studies to obtain a clearer picture of the relationship between fruit juice consumption and blood lipid levels in humans.

## 6. Conclusions

There is ample evidence that links sugar-sweetened beverage consumption to adverse effects on metabolic risk factors, such as dyslipidemia [35,41,42,58,59,60,61,62,63,64,65] and insulin resistance [35,41,42,58,61,65]. Despite the similarity of fruit juices to sugar-sweetened beverages in terms of free sugar and fructose content, it remains unclear in the current literature whether they lead to the same metabolic consequences if consumed in equal doses. More randomized controlled trials comparing the metabolic effects of fruit juice and sugar-sweetened beverage consumption under dietary conditions that will eliminate confounding differences due to diet are needed in order to shape clear evidence-based public health guidelines. These future studies should also address important aspects such as individuals’ gene variants, metabolic status, medical treatment, and physical activity levels, but also more qualitative characteristics such as eating habits and behaviors. Most consumers do not ingest as much fruit juice as sugar-sweetened beverages. Indeed, a systematic assessment of beverage intake (from annual food balance information) has estimated the average daily consumption of fruit juices in American adults over 20 years old in 2010 to be approximately 80 mL and the consumption of sugar-sweetened beverages to be 237 mL [100]. Would it be beneficial to their health if Americans reversed this pattern and consumed 237 mL/day of fruit juices and only 80 mL of sugar-sweetened beverages? The question still needs to be answered: Are fruit juices healthier than sugar-sweetened beverages?

## Figures and Tables

**Figure 1 nutrients-11-01006-f001:**
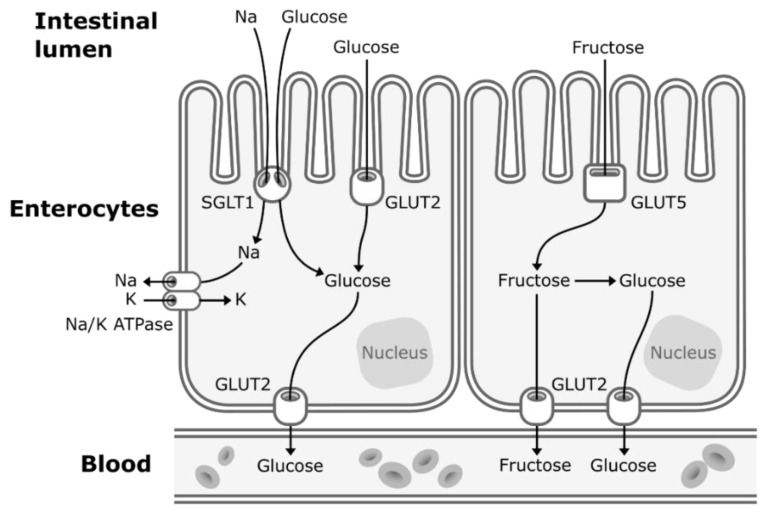
Absorption of fructose and glucose in the enterocytes. Glucose enters the enterocytes mostly by secondary active transport via sodium-glucose transporters (SGLT1) located in the apical membrane of the enterocytes. Under high concentrations of glucose in the lumen of the intestine, glucose also enters the enterocytes by facilitated diffusion via glucose transporters (GLUT2). Fructose enters the enterocytes through a specific fructose transporter (GLUT5). Then, both glucose and fructose exit the enterocytes to enter the systematic circulation by facilitated diffusion via GLUT2 transporters located in the basolateral membrane of the enterocytes. A small part of dietary fructose will be converted and released in the bloodstream by the enterocytes as glucose.

**Figure 2 nutrients-11-01006-f002:**
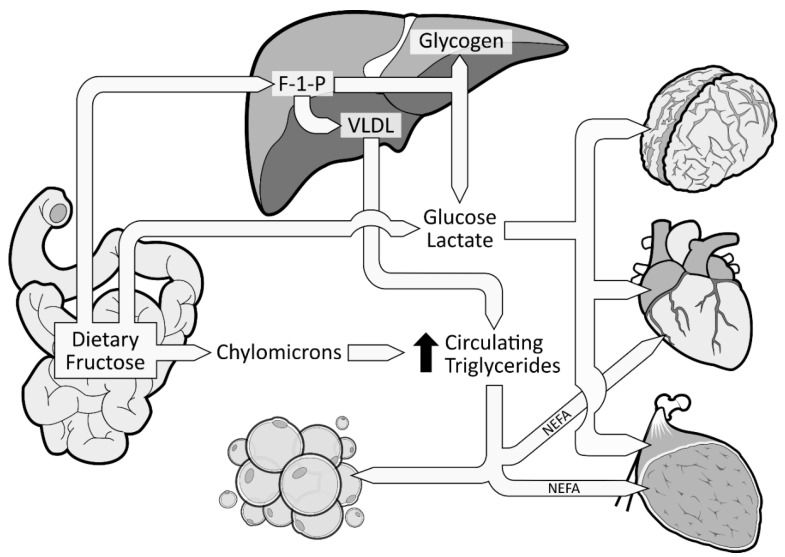
Metabolic fate of dietary fructose. Dietary fructose is ingested and released by the enterocytes mostly as fructose but also converted and released as glucose, lactate, and fatty acids (in chylomicrons). Fructose spills over to the liver where it is phosphorylated as Fructose 1-Phosphate (F 1-P). The largest part of F 1-P will be metabolized and converted by the hepatocytes as glucose, which can be stored as glycogen or released in the bloodstream [35]. Hepatocytes can also convert F 1-P into lactate and fatty acids. Fatty acids accumulate into the liver, consequently favoring the production and secretion of very low-density lipoproteins (VLDL), which leads to increased levels of circulating triglycerides and dyslipidemia.

**Figure 3 nutrients-11-01006-f003:**
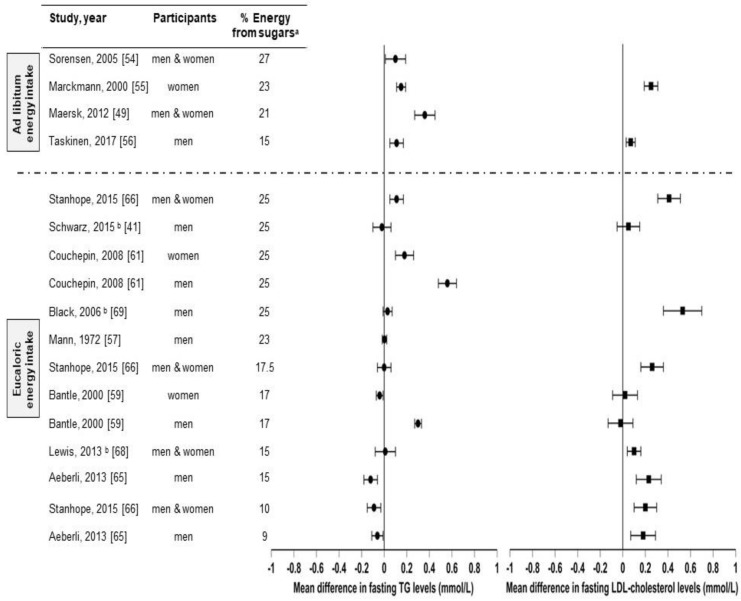
Effects of sugar consumption from sugar-sweetened beverages on fasting blood triglyceride (TG) and fasting LDL-cholesterol levels in healthy individuals with normal weight, overweight, or obesity. Mean difference in fasting blood triglyceride and fasting low-density lipoprotein cholesterol (LDL-C) levels in studies that compared higher with lower sugar intakes from sugar-sweetened beverages in healthy individuals with normal weight, overweight, or obesity. ^a^ Refers to the higher sugar intake intervention. The percentage of energy from the lower sugar intake intervention is detailed in Table I. Studies with ad libitum energy intake controlled for a minimal sugar intake but not for total energy intake. Studies with eucaloric energy intake controlled for a minimal sugar intake and for weight maintenance throughout the studies. ^b^ Studies that controlled for a minimal sugar intake, for weight maintenance throughout the studies, and for diet macronutrient variations between experimental groups.

**Table 1 nutrients-11-01006-t001:** Characteristics of studies included in Figure 3. Detailed characteristics of studies included in Figure 3 in terms of participants (sample size, sex, weight, age), duration, intervention (% Ereq of sugar intake provided in sugar-sweetened beverages only, sugar-sweetened beverages and food, or unspecified), and control (comparative: low-sugar intervention or baseline values).

	First Author, Year of Publication [ref]	Participants	Duration	Intervention	Control
Ad libitum energy intake with usual diets	Sorensen, 2005 [54]	6 healthy men with overweight 35 healthy women with overweight Age: 24–47 year	10 weeks	27% Ereq sucrose; Sweetened beverages and food	4% sucrose; Food
	Marckmann, 2000 [55]	20 healthy women with normal weight Age: 21–52 year	2 weeks	23% Ereq sucrose; Sweetened beverages and food	3% Ereq sucrose; Sweetened beverages and food
	Maersk, 2012 [49]	17 healthy men with overweight 30 healthy women with overweight Age: 33–45 year	6 months	21% Ereq sucrose; Sweetened beverages (1000 mL)	Compared to baseline values
	Taskinen, 2017 [56]	82 healthy men with obesity Age: 20–65 year	12 weeks	15% Ereq fructose; Sweetened beverages (3 × 330 mL)	Compared to baseline values
Eucaloric energy intake with isocaloric diets	Stanhope, 2015 [66]	42 healthy men with normal weight to obesity 43 healthy women with normal weight to obesity Age:18–40 year	19 days	10% Ereq HFCS 17.5% Ereq HFCS 25% Ereq HFCS; Sweetened beverages	Artificially sweetened beverages (aspartame)
	Schwarz, 2015 [41]	8 healthy men with normal weight to obesity Age: 18–65 year	9 days	25% Ereq fructose; Sweetened beverages	Isocaloric substitution of fructose for complex carbohydrates; Sweetened beverages
	Couchepin, 2008 [61]	8 healthy men with normal weight 8 healthy women with normal weight Age: 21–23 year	2 × 6 days 4 weeks washout	25% Ereq fructose; (Eucaloric low fructose diet + overfeeding with 25% Ereq fructose); Unspecified medium	Eucaloric low fructose diet; Unspecified medium
	Black, 2006 [69]	13 healthy men with normal weight Age: 30–36 year	2 × 6 weeks unspecified washout period	25% Ereq sucrose; Sweetened beverages and food	10% Ereq sucrose; Sweetened beverages and food
	Mann, 1972 [57]	9 healthy men with normal weight Age: 30–40 year	3 × 14 days	23% Ereq sucrose; Unspecified medium	Isocaloric substitution of sucrose for complex carbohydrates; Unspecified medium
	Bantle, 2000 [59]	12 healthy men with normal weight 12 healthy women with normal weight Age: <40 year (6 men, 6 women) >40 year (6 men, 6 women)	2 × 6 weeks 4 weeks washout	17% Ereq fructose; Sweetened beverages and food	17% Ereq glucose; Sweetened beverages and food
	Lewis, 2013 [68]	13 healthy men and women with overweight or obesity Age: 35–56 year	2 × 6 weeks 4 weeks washout	15% Ereq sucrose; Sweetened beverages and food	5% Ereq sucrose; Sweetened beverages and food
	Aeberli, 2013 [65]	9 healthy men with normal weight Age: 21–25 year	4 × 3 weeks 4 weeks washout	9% Ereq fructose 14% Ereq fructose 15% Ereq sucrose; Sweetened beverages (3 × 200 mL)	15% Ereq glucose; Sweetened beverages (3 × 200 mL)

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
