# Peer review of "Are Fruit Juices Healthier Than Sugar-Sweetened Beverages? A Review"

_nutrients, 2019, doi:10.3390/nu11051006_

Round 1
Reviewer 1 Report
Thank you for the opportunity to review this manuscript.
I really enjoyed reading this manuscript.
Please display Table 1 widthwise for readerbility.
2. The reference number have to move end of the sentence (line 398).
Author Response
We would like to thank the reviewer for their examination and understanding of our manuscript.

Reviewer 2 Report
Line 72 - it should be noted that many are changing from the use of HFCS
Ref 14 is quite old and does not reflect the decrease in consumption.
Line 74 - be sure to emphasize that these are past data as consumption of HCFS and SSB has changed
Line 95 ditto and you should not use company names – it should be major brands of cola beverages.
Line 116-119 Ref 22 says nothing about the amount of the free sugars
Is there data on the increased use of HFCS 65 – this seems like a guesstimate
Line 208 – 209 – It would not matter if it were HCFS or sucrose in an acidic medium – sucrose would invert.
Line 288 – It seems that you should note while the main difference in free sugars – it is important to note that fruit juice has as its main component sugars that the phytochemicals and nutrients make a big difference
lines 324-391 should talk about the fact that consumers may not consume amounts of juice that are equivalent to the overconsumption of SSB ; consumption data would be useful and 250 ml is probably the most that the bulk of consumers would ingest. I think it is not common for consumers to ingest 750 ml of OJ and while interesting may have little bearing in the real world.
Line 369-70 points out differences in background diets but also should point out that it is unlikely that if randomization were well carried out that the differences in diet alluded to would be unlikely.
Line 396 - the study should talk about SSBs and fruit juices – it did not look at fructose in other processed foods, so no mention should be made of them
Overall the work does not give a great deal of clarity on the issue. It appears that a nearly equal number of studies say that it increases risks of some health points as find no effect or benefits. It could be that background diet makes an incredible difference as does characteristics of the eater and other aspects that are not addressed.
Author Response
We would like to thank the reviewer for their thorough examination and understanding of our manuscript. Please see a point-by-point response to the reviewer's comments in attached file.
